# Neutron Star Binaries Produced by Binary-Driven Hypernovae, Their Mergers, and the Link between Long and Short GRBs

Laura M. Becerra [1,2,*,†], Chris Fryer [3,†], Jose F. Rodriguez [1,2,*,†], Jorge A. Rueda [2,4,5,6,7*,†] and Remo. Ruffini [2,4,8,†]

1    *GIRG*, Escuela de Física, Universidad Industrial de Santander, Bucaramanga 680002, Colombia
2    ICRANet, Piazza della Repubblica 10, 65122 Pescara, Italy
3    CCS-2, Los Alamos National Laboratory, Los Alamos, NM 87545, USA
4    ICRA, Dip. di Fisica, Sapienza Università di Roma, P.le Aldo Moro 5, 00185 Rome, Italy
5    ICRANet-Ferrara, Dip. di Fisica e Scienze Della Terra, Università Degli Studi di Ferrara, Via Saragat 1, 44122 Ferrara, Italy
6    Dip. di Fisica e Scienze della Terra, Università degli Studi di Ferrara, Via Saragat 1, 44122 Ferrara, Italy
7    INAF, Istituto di Astrofisica e Planetologia Spaziali, Via Fosso del Cavaliere 100, 00133 Rome, Italy
8    INAF, Viale del Parco Mellini 84, 00136 Rome, Italy
*    Correspondence: laura.becerra7@correo.uis.edu.co (L.M.B.); joferoru@gmail.com (J.F.R.); jorge.rueda@icra.it (J.A.R.)
†    These authors contributed equally to this work.

**Abstract:** The binary-driven hypernova (BdHN) model explains long gamma-ray bursts (GRBs) associated with supernovae (SNe) Ic through physical episodes that occur in a binary composed of a carbon-oxygen (CO) star and a neutron star (NS) companion in close orbit. The CO core collapse triggers the cataclysmic event, originating the SN and a newborn NS (hereafter $\nu$NS) at its center. The $\nu$NS and the NS accrete SN matter. BdHNe are classified based on the NS companion fate and the GRB energetics, mainly determined by the orbital period. In BdHNe I, the orbital period is of a few minutes, so the accretion causes the NS to collapse into a Kerr black hole (BH), explaining GRBs of energies $>10^{52}$ erg. BdHN II, with longer periods of tens of minutes, yields a more massive but stable NS, accounting for GRBs of $10^{50}$–$10^{52}$ erg. BdHNe III have still longer orbital periods (e.g., hours), so the NS companion has a negligible role, which explains GRBs with a lower energy release of $<10^{50}$ erg. BdHN I and II might remain bound after the SN, so they could form NS-BH and binary NS (BNS), respectively. In BdHN III, the SN likely disrupts the system. We perform numerical simulations of BdHN II to compute the characteristic parameters of the BNS left by them, their mergers, and the associated short GRBs. We obtain the mass of the central remnant, whether it is likely to be a massive NS or a BH, the conditions for disk formation and its mass, and the event's energy release. The role of the NS nuclear equation of state is outlined.

**Keywords:** neutron stars; gamma-ray burst; close binaries

## 1. Introduction

Gamma-ray bursts (GRBs) are classified using the time (in the observer's frame) $T_{90}$, in which 90% of the observed isotropic energy ($E_{iso}$) in the gamma-rays is released. Long GRBs have $T_{90} > 2$ s and short GRBs, $T_{90} < 2$ s [1–5]. The two types of sources, short and long GRBs, are thought to be related to phenomena occurring in gravitationally collapsed objects, e.g., stellar-mass black holes (BHs) and neutron stars (NSs).

For short GRBs, mergers of binary NSs (BNSs) and/or NS-BH were soon proposed as progenitors [6–9]). For long bursts, the core-collapse of a single massive star leading to a BH (or a magnetar), a *collapsar* [10], surrounded by a massive accretion disk has been the traditional progenitor (see, e.g., [11,12], for reviews). The alternative binary-driven hypernova (BdHN) model exploits the increasing evidence for the relevance of a binary

progenitor for long GRBs, e.g., their association with Ic-type supernovae (SNe) [13–16], proposing a binary system composed of a carbon-oxygen star (CO) and an NS companion for long GRBs. We refer the reader to [17–23] for theoretical details on the model.

In this article, we are interested in the direct relationship between long and short GRBs predicted by the BdHN scenario. The CO undergoes core collapse, ejecting matter in a supernova (SN) explosion and forming a newborn NS ($\nu$NS) at its center. The NS companion attracts part of the ejected material leading to an accretion process with high infalling rates. Also, the $\nu$NS gains mass via a fallback accretion process. The orbital period is the most relevant parameter for the CO-NS system's fate. In BdHN of type I, the NS reaches the critical mass, gravitationally collapsing into a Kerr BH. It occurs for short orbital periods (usually a few minutes) and explains GRBs with energies above $10^{52}$ erg. In BdHN II, the orbital period is larger, up to a few tens of minutes, so the accretion rate decreases, and the NS becomes more massive but remains stable. These systems explain GRBs with energies $10^{50}$–$10^{52}$ erg. In BdHN III, the orbital separation is still larger; the NS companion does not play any role, and the energy release is lower than $10^{50}$ erg. If the binary is not disrupted by the mass loss in the SN explosion (see [20] for details), a BdHN I produces a BH-NS, whereas a BdHN II produces a BNS. In BdHN III, the SN is expected to disrupt the system. Therefore, in due time, the mergers of NS-BHs left by BdHNe I and of BNS left by BdHNe II are expected to lead to short GRBs.

Short GRBs from BNS mergers have been classified into short gamma-ray flashes (S-GRFs) and authentic short GRBs (S-GRBs), depending on whether the central remnant is an NS or a BH, respectively [24]. Two different subclasses of short GRBs from BNS mergers have been electromagnetically proposed [20,24,25]:

(1) *Authentic short GRBs (S-GRBs)*: short bursts with isotropic energy $E_{\rm iso} \gtrsim 10^{52}$ erg and peak energy $E_{p,i} \gtrsim 2$ MeV. They occur when a BH is formed in the merger, which is revealed by the onset of a GeV emission (see [25–27]). Their electromagnetically inferred isotropic occurrence rate is $\rho_{\rm S-GRB} \approx \left(1.9^{+1.8}_{-1.1}\right) \times 10^{-3}$ Gpc$^{-3}$ year$^{-1}$ [24]. The distinct signature of the formation of the BH, namely the observation of the 0.1–100 GeV emission by the *Fermi*-LAT, needs the presence of baryonic matter interacting with the newly-formed BH, e.g., via an accretion process (see, e.g., [26,28]).

(2) *Short gamma-ray flashes (S-GRFs)*: short bursts with $E_{\rm iso} \lesssim 10^{52}$ erg and $E_{p,i} \lesssim 2$ MeV. They occur when no BH is formed in the merger, i.e., when it leads to a massive NS. Their U-GRB electromagnetically inferred isotropic occurrence rate is $\rho_{\rm S-GRF} \approx 3.6^{+1.4}_{-1.0}$ Gpc$^{-3}$ year$^{-1}$ [24].

(3) *Ultrashort gamma-ray flashes (U-GRFs)*: in [20], it has been advanced a new class short bursts, the ultrashort GRBs (U-GRBs) produced by NS-BH binaries when the merger leaves the central BH with very little or completely without surrounding matter. An analogous system could be produced in BNS mergers. We shall call these systems ultrashort GRFs, for short U-GRFs. Their gamma-ray emission is expected to occur in a prompt short radiation phase. The post-merger radiation is drastically reduced, given the absence of baryonic matter to power an extended emission. A *kilonova* can still be observed days after the merger, in the infrared, optical, and ultraviolet wavelengths, produced by the radioactive decay of r-process yields [29–32]. Kilonova models used a *dynamical* ejecta composed of matter expelled by tides prior or during the merger, and a *disk-wind* ejecta by matter expelled from post-merger outflows in accretion disks [33], so U-GRFs are expected to have only the dynamical ejecta kilonova emission.

We focus on the BNSs left by BdHNe II and discuss how their properties impact the subsequent merger process and the associated short GRB emission, including their GW radiation. Since an accretion disk around the central remnant of a BNS merger, i.e., a newborn NS or a BH, is an important ingredient in models of short GRBs (see, e.g., [34] and references therein), we give some emphasis to the conditions and consequences for the merger leaving a disk. We study BNSs formed through binary evolution channels. Specifically, we expect these systems to form following a binary evolution channel similar to that of two massive stars leading to stripped-envelope binaries, described in previous studies (e.g., [35,36]). In this process, the CO star undergoes mass loss in multiple mass-

transfer and common-envelope phases through interactions with the NS companion (see, e.g., [37–39]). This leads to removing the H/He layers of the secondary star, which ends up as a CO star. Recently, significant progress has been made in the study of alternative evolution channels for the progenitor of BNSs, such as hierarchical systems involving triple and quadrupole configurations [40,41], which are motivated by the presence of massive stars in multiple systems [42]. These systems are out of the scope of this study.

The article is organized as follows. In Section 2, we discuss the numerical simulations of BdHNe and specialize in an example of a BNS led by a BdHN II. Section 3 introduces a theoretical framework to analyze the BNS merger outcome configuration properties based on the conservation laws of baryon number, angular momentum, and mass-energy. We present in Section 4 a specific example analyzing a BNS merger using the above-mentioned theoretical framework, including estimates of the energy and angular momentum release. We include the radiation in gravitational waves (GWs) and estimate its detection by current facilities. Section 5 presents a summary and the conclusions of this work.

## 2. A BNS Left by a BdHN II

Figure 1 shows a snapshot of the mass density with the vector velocity field at the binary's equatorial plane some minutes after the CO collapse and the expansion of the SN ejecta. The system's evolution was simulated with an SPH code, where the NS companion and the $\nu$NS are point particles that interact gravitationally with the SPH particles of the SN ejecta. For details of these numerical simulations, we refer to [23,43]. In these simulations, the influence of the star's magnetic field has be disregarded, as the magnetic pressure remains significantly lower than the random pressure exerted on the infalling material. The simulation of Figure 1 corresponds to a CO-NS for a CO star evolved from a zero-age main-sequence (ZAMS) star of $M_{zams} = 15\,M_\odot$. The CO mass is about $3.06\,M_\odot$, whose core collapse leaves a $1.4\,M_\odot$ $\nu$NS and ejects $1.66\,M_\odot$. The NS companion's initial mass is $1.4\,M_\odot$, and the initial binary period of the system is about 4.5 min.

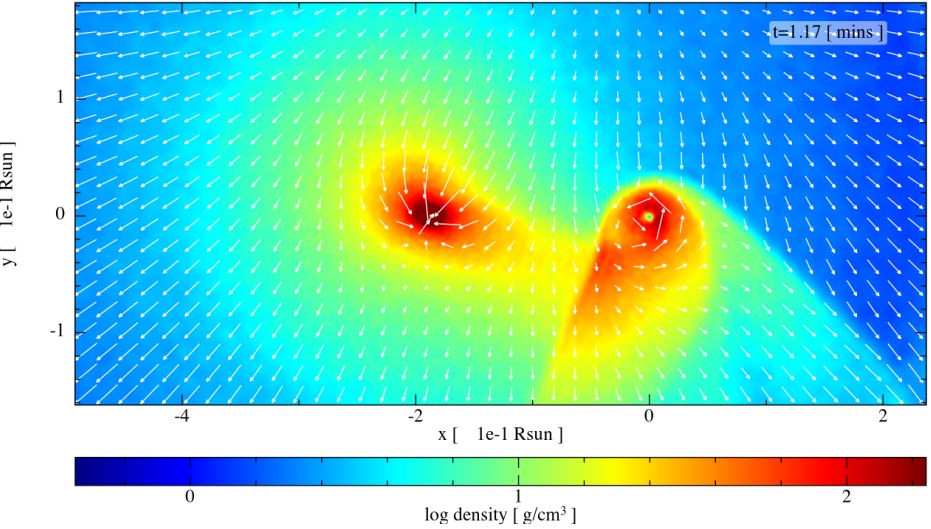

**Figure 1.** Massdensity snapshots and velocity field on the orbital plane of a BdHN for a CO left by a $M_{zams} = 15\,M_\odot$ and a $1.4\,M_\odot$ NS companion, with an initial orbital period of about 4.5 min. We follow the expansion of the SN ejecta in the presence of the NS companion and the $\nu$−NS with a smoothed particle hydrodynamic (SPH) code. It is clear that a disk with opposite spins has formed around both stars.

From the accretion rate on the NSs, we have calculated the evolution of the mass and angular momentum of the binary components (see [43], for details). Table 1 summarizes the final parameters of the $\nu$NS and the NS, including the gravitational mass, *m*, dimensionless angular momentum, *j*, angular velocity, $\Omega$, equatorial radius, $R_{eq}$, and moment of inertia,

*I*. These structure parameters have been calculated with the RNS code [44] and using the GM1 [45,46] and TM1 [47] EOS (see Table 2 for details of the EOS). The BNS left by the BdHN II event has a period $P_{orb} = 14.97$ min, orbital separation $a_{orb} \approx 2 \times 10^{10}$ cm, and eccentricity $e = 0.45$.

**Table 1.** BNS produced by a BdHN II originated in a CO-NS with an orbital period of 4.5 min. The CO star mass is 3.06 $M_\odot$, obtained from the stellar evolution of a ZAMS star of $M_{zams} = 15\ M_\odot$, and the NS companion has 1.4 $M_\odot$. The numerical smoothed-particle hydrodynamic (SPH) simulation follows the SN produced by the CO core collapse and estimates the accretion rate onto the $\nu$NS and the NS companion. The structure parameters of the NSs are calculated for the GM1 and TM1 EOS. We refer to [43] for additional details.

| | $m$ [$M_\odot$] | $j$ | $\Omega$ [$s^{-1}$] | $R_{eq}$ [km] | $I$ [g cm$^2$] | $\Omega$ [$s^{-1}$] | $R_{eq}$ [km] | $I$ [g cm$^2$] |
|---|---|---|---|---|---|---|---|---|
| | | | | **GM1 EOS** | | | **TM1 EOS** | |
| $\nu$NS | 1.505 | 0.259 | 1114.6 | 14.03 | $2.04 \times 10^{45}$ | 1077.1 | 14.47 | $2.11 \times 10^{45}$ |
| NS | 1.404 | $-0.011$ | $-52.14$ | 14.01 | $1.85 \times 10^{45}$ | $-56.6$ | 14.49 | $1.93 \times 10^{45}$ |

**Table 2.** Properties of the selected EOS. From left to right: maximum stable mass of non-rotating configurations, uniformly rotating configurations, set by the maximum mass of the Keplerian/mass-shedding sequence and the corresponding angular velocity.

| EOS | $M_{max}^{j=0}$ [$M_\odot$] | $M_{max}^{j_{kep}}$ [$M_\odot$] | $\Omega_{kep}^{max}$ [$s^{-1}$] |
|---|---|---|---|
| GM1 | 2.38 | 2.84 | $1.001 \times 10^4$ |
| TM1 | 2.19 | 2.62 | $8.83 \times 10^3$ |

## 3. Inferences from Conservation Laws

We analyze the properties of the central remnant NS formed after the merger. We use the conservation laws of baryon number, energy, and angular momentum for this aim.

### 3.1. Baryon Number Conservation

The total baryonic mass of the system must be conserved, so the binary baryonic mass, $M_b$, will redistribute among that of the postmerger's central remnant, $m_{b,c}$; the ejecta's mass, $m_{ej}$, which is unbound to the system; and the matter kept bound to the system, e.g., in the form of a disk of mass $m_d$. Therefore, we have the constraint

$$M_b = m_{b,c} + m_{ej} + m_d, \quad M_b = m_{b,1} + m_{b,2}. \tag{1}$$

For a uniformly rotating NS, the relation among its baryonic mass, $m_{b,i}$, gravitational mass, $m_i$, and angular momentum $J_i$, is well represented by the simple function

$$\frac{m_{b,i}}{M_\odot} \approx \frac{m_i}{M_\odot} + \frac{13}{200} \left( \frac{m_i}{M_\odot} \right)^2 \left( 1 - \frac{1}{130} j_i^{1.7} \right), \quad i = 1, 2, c, \tag{2}$$

where $j_i \equiv cJ_i/(GM_\odot^2)$, which fits numerical integration solutions of the axisymmetric Einstein equations for various nuclear EOS, with a maximum error of 2% [48]. Thus, Equation (2) is a nearly universal, i.e., EOS-independent, formula. Equation (2) applies to the merging components ($i = 1, 2$) as well as to the central remnant ($i = c$).

### 3.2. Angular Momentum Conservation

We can make more inferences about the merger's fate from the conservation of angular momentum. The angular momentum of the binary during the inspiral phase is given by

$$J = \mu r^2 \Omega + J_1 + J_2, \quad J_i = \frac{2}{5}\kappa_i m_i R_i^2 \Omega_i, \quad i = 1, 2, \tag{3}$$

where $r$ is the orbital separation, $\mu = m_1 m_2 / M$ is the reduced mass, $M = m_1 + m_2$ is the total binary mass, and $\Omega = \sqrt{GM/r^3}$ is the orbital angular velocity. The gravitational mass and stellar radius of the $i$-th stellar component are, respectively, $m_i$ and $R_i$; $J_i$ is its angular momentum, $\Omega_i$ its angular velocity, and $\kappa_i$ is the ratio between its moment of inertia to that of a homogeneous sphere. We adopt the convention $m_2 \leq m_1$. After the merger, the angular momentum is given by the sum of the angular momentum of the central remnant, the disk, and the ejecta. Angular momenta conservation implies that the angular momenta at merger, $J_{\mathrm{merger}}$, equals that of the final configuration plus losses:

$$J_{\mathrm{merger}} = J_c + J_d + \Delta J, \tag{4}$$

where $J_c$ and $J_d$ are, respectively, the angular momenta of the central remnant and the eventual surrounding disk, $\Delta J$ accounts for angular momentum losses, e.g., via gravitational waves, and we have neglected the angular momentum carried out by the ejecta since it is expected to have small mass $\sim 10^{-4}$–$10^{-2} M_\odot$. Simulations suggest that this ejecta comes from interface of the merger, where matter is squeezed and ejected perpendicular to the orbital plane, see, e.g., [49,50]. The definition of the merger point will be discussed below.

The angular momentum of the binary at the merger point is larger than the maximum value a uniformly rotating NS can attain, i.e., the angular momentum at the Keplerian/mass-shedding limit, $J_K$. Thus, the remnant NS should evolve first through a short-lived phase that radiates the extra angular momentum over that limit and enters the rigidly rotating stability phase from the mass-shedding limit. Thus, we assume the remnant NS after that transition phase starts its evolution with angular momentum

$$J_c = J_K \approx 0.7 \frac{G m_c^2}{c}. \tag{5}$$

Equation (5) fits the angular momentum of the Keplerian sequence from full numerical integration of the Einstein equations and is nearly independent of the nuclear EOS (see, e.g., [48] and references therein). Therefore, the initial dimensionless angular momentum of the central remnant is

$$j_c = \frac{c J_c}{G M_\odot^2} \approx 0.7 \left( \frac{m_c}{M_\odot} \right)^2. \tag{6}$$

We model the disk's angular momentum as a ring at the remnant's inner-most stable circular orbit (ISCO). Thus, we use the formula derived in Cipolletta et al. [51], which fits, with a maximum error of 0.3%, the numerical results of the angular momentum per unit mass of a test particle circular orbit in the general relativistic axisymmetric field of a rotating NS. Within this assumption, the disk's angular momentum is given by

$$J_d = J_{\mathrm{ISCO}} \approx \frac{G}{c} m_c m_d \left[ 2\sqrt{3} - 0.37 \left( \frac{j_c}{m_c/M_\odot} \right)^{0.85} \right]. \tag{7}$$

Notice that Equation (7) reduces to the known result for the Schwarzschild metric for vanishing angular momentum, as it must. However, it differs from the result for the Kerr metric, which tells us that the Kerr metric does not describe the exterior spacetime of a rotating NS (see [51] for a detailed discussion).

The estimate of $J_{\mathrm{merger}}$ requires the knowledge of the merger point, which depends on whether or not the binary secondary becomes noticeably deformed by the tidal forces.

When the binary mass ratio $q \equiv m_2/m_1$ is close or equal to 1, the stars are only deformed before the point of contact [52]. Therefore, for $q \approx 1$, we can assume the point of the merger as the point of contact

$$r_{\text{merger}} \approx r_{\text{cont}} = \frac{(\mathcal{C}_2 + q\mathcal{C}_1)}{(1+q)\mathcal{C}_1\mathcal{C}_2} \frac{GM}{c^2}, \tag{8}$$

where $\mathcal{C}_{1,2} \equiv Gm_{1,2}/(c^2 R_{1,2})$ is the compactness of the BNS components.

When the masses are different, if we model the stars as Newtonian incompressible spheroids, there is a minimal orbital separation $r_{\text{ms}}$, below which no equilibrium configuration is attainable, i.e., one star begins to shed mass to the companion due to the tidal forces. In this approximation, $r_{\text{ms}} \approx 2.2q^{-1/3}R_2$ [53]. Numerical relativity simulations of BH-NS quasi-equilibrium states suggest that the mass-shedding occurs at a distance (see [54] and references therein) of

$$r_{\text{ms}} \approx (0.270)^{-2/3}q^{-1/3}R_2. \tag{9}$$

Our analysis adopts the mass-shedding distance of Equation (9). For a system with $q = 0.7$ (similar mass ratio of the one in Table 1), we have found that the less-compact star begins to shed mass before the point of contact, independently of the EOS, which agrees with numerical relativity simulations. Consequently, for non-symmetric binaries $q < 1$, we define the merging at the point as the onset of mass-shedding, $r_{\text{merger}} \approx r_{\text{ms}}$.

Based on the above two definitions of merger point, Equations (8) and (9), the angular momentum at the merger is given by

$$J_{\text{merger}} = \begin{cases} \nu\sqrt{\frac{\mathcal{C}_2 + q\mathcal{C}_1}{(1+q)\mathcal{C}_1\mathcal{C}_2}} \frac{GM^2}{c}, & q \approx 1, \\ \nu q^{1/3}[(1+q)\mathcal{C}_2]^{-1/2} \frac{GM^2}{c}, & q < 1, \end{cases} \tag{10}$$

where we have introduced the so-called symmetric mass-ratio parameter, $\nu \equiv q/(1+q)^2$.

### 3.3. Mass-Energy Conservation

The conservation of mass-energy before and after the merger implies the energy released equals the mass defect of the system, i.e.,

$$E_{\text{GW}} + E_{\text{other}} = \Delta M c^2 = [M - (m_c + m_{\text{ej}} + m_d)]c^2, \tag{11}$$

where $\Delta M$ is the system's mass defect. We have also defined $E_{\text{GW}} = E_{\text{GW}}^{\text{insp}} + E_{\text{GW}}^{\text{pm}}$ the total energy emitted in GWs in the inspiral regime, $E_{\text{GW}}^{\text{insp}}$, and in the merger and post-merger phases, $E_{\text{GW}}^{\text{pm}}$. The energy $E_{\text{other}}$ is radiated in channels different from the GW emission, e.g., electromagnetic (photons) and neutrinos.

## 4. A Specific Example of BNS Merger

We analyze the merger of the $1.505 + 1.404\ M_\odot$ BNS in Table 1. For these component masses, the inferred orbital separation of $a_{\text{orb}} \approx 2 \times 10^{10}$ cm and eccentricity $e = 0.45$, the merger is expected to be driven by GW radiation on a timescale [55] of

$$\tau_{\text{GW}} = \frac{c^5}{G^3} \frac{5}{256} \frac{a_{\text{orb}}^4}{\mu M^2} F(e) \approx 73.15 \text{ kyr}, \quad F(e) = \frac{48}{19} \frac{1}{g(e)^4} \int_0^e \frac{g(e)^4(1-e^2)^{5/2}}{e(1+\frac{121}{304}e^2)} de \approx 0.44, \tag{12}$$

where $g(e) = e^{12/19}(1-e^2)^{-1}(1+121e^2/304)^{870/2299}$.

From Equations (2), (7) and (10), and the conservation Equations (1), (4) and (11), we can obtain the remnant and disk's mass as a function of the angular momentum losses, $\Delta J$, as well as an estimate of the energy and angular momentum released in the cataclysmic event. We use the NS structure parameters obtained for the GM1 EOS and the TM1 EOS. The total gravitational mass of the system is $M = m_1 + m_2 = 2.909\ M_\odot$, so using Equation (2), we obtain the total baryonic mass of the binary, $M_b = m_{b,1} + m_{b,2} \approx 3.184\ M_\odot$. The binary's mass fraction is $q = 0.933$, so we assume the merger starts at the contact point. With this,

the angular momentum at the merger, as given by Equation (10), for the GM1 and TM1 EOS is, respectively, $J_{\text{merger}} \approx 5.65\ GM_\odot^2/c$ and $J_{\text{merger}} \approx 5.73\ GM_\odot^2/c$.

Figure 2 shows that the disk's mass versus the central remnant's mass for selected values of the angular momentum loss for the two EOS. The figure shows the system's final parameters lie between two limiting cases: zero angular momentum loss leading to maximal disk mass and maximal angular momentum loss leading to zero disk mass.

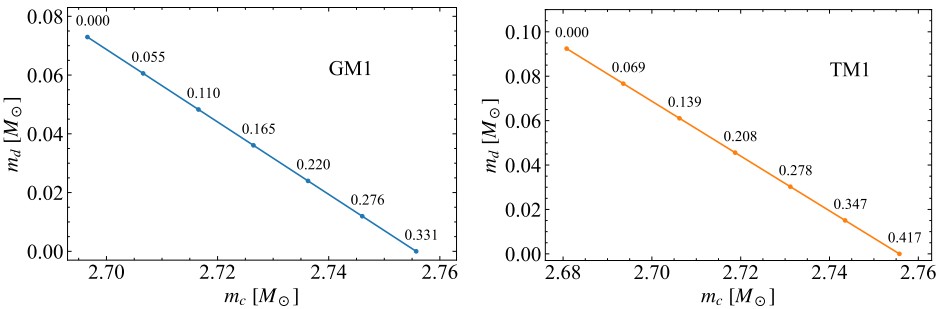

**Figure 2.** Disk mass versus central remnant (NS) mass. Selected values of the angular momentum loss (in units of $GM_\odot^2/c$) are shown as points. The initial BNS has a total gravitational mass of 2.909 $M_\odot$ and a mass fraction $q = 0.933$, so we assume the merger starts at the contact point. The maximum mass along the Keplerian sequence for the GM1 EOS is 2.84 $M_\odot$ and for the TM1 EOS it is 2.62 $M_\odot$ (see Table 2). Thus, for the former EOS, the central remnant is a massive fast-rotating NS, while the latter suggests a prompt collapse into a Kerr BH.

*4.1. Maximal Disk Mass*

We obtain the configuration corresponding to the maximum disk mass switching off angular momentum losses. Let us specialize in the GM1 EOS. By setting $\Delta J = 0$, the solution of the system of equations formed by the baryon number and angular conservation equations leads to the central remnant's mass, $m_c = 2.697\ M_\odot$, and disk's mass, $m_d = 0.073\ M_\odot$. This limiting case switches off the GW emission, so it also sets an upper limit to the energy released in mechanisms different than GWs. Thus, Equation (11) implies that $E_{\text{other}} = \Delta Mc^2 = [M - (m_c + m_{\text{ej}} + m_d)]c^2 \approx (M - m_c - m_d)c^2 \approx 0.139\ M_\odot c^2 \approx 2.484 \times 10^{53}$ erg of energy are carried out to infinity by a mechanism different than GWs and not accompanied by angular momentum losses.

*4.2. Zero Disk Mass*

The other limiting case corresponds when the angular momentum loss and the remnant mass are maximized, i.e., when no disk is formed (see Figure 2). By setting $m_d = 0$, the solution of the conservation equations leads to the maximum angular momentum loss, $\Delta J = 0.331\ GM_\odot^2/c$, and the maximum remnant's mass, $m_c = 2.756\ M_\odot$.

Thus, the upper limit to the angular momentum carried out by GWs is given by the maximum amount of angular momentum losses, i.e., $\Delta J_{\text{GW}} \lesssim 0.331\ GM_\odot^2/c$. In the inspiral phase of the merger, the system releases

$$E_{\text{GW}}^{\text{insp}} \approx \frac{Gm_1 m_2}{2r_{\text{cont}}} = \frac{q\mathcal{C}_1\mathcal{C}_2 Mc^2}{2(1+q)(\mathcal{C}_2 + q\mathcal{C}_1)} + \frac{1}{2}\left[j_1|\Omega_1| + j_2|\Omega_2|\right]\frac{GM_\odot^2}{c}. \qquad (13)$$

For the binary we are analyzing, $E_{\text{GW}}^{\text{insp}} \approx 0.0194\ Mc^2 \approx 0.0563 M_\odot c^2 \approx 1.0073 \times 10^{53}$ erg. The transitional non-axisymmetric object (e.g., triaxial ellipsoid) formed immediately after the merger mainly generates these GWs, and their emission ends when the stable remnant NS is finally formed. We can model such a rotating object as a compressible ellipsoid with a polytropic EOS of index $n = 0.5$–1 [56]. The object will spin up by angular momentum loss to typical frequencies of 1.4–2.0 kHz. The energy emitted in GWs is $E_{\text{GW}}^{\text{pm}} \approx 0.0079\ M_\odot c^2 \approx 1.404 \times 10^{52}$ erg. Therefore, the energy released in GWs is,

$E_{GW} = E_{GW}^{insp} + E_{GW}^{pm} \approx 0.0642\, M_\odot c^2 \approx 1.147 \times 10^{53}$ erg. If no disk is formed, i.e., for a U-GRF, the mass-energy defect is $\Delta M c^2 = [M - (m_c + m_{ej})]c^2 \approx (M - m_c)c^2 \approx 0.153\, M_\odot c^2 \approx 2.734 \times 10^{53}$ erg. This implies that $E_{other} = \Delta M c^2 - E_{GW} \approx 0.089\, M_\odot c^2 \approx 1.591 \times 10^{53}$ erg are released in forms of energy different than GW radiation.

Therefore, combining the above two results, we conclude that for the present merger, assuming the GM1 EOS, the merger releases $0 < E_{GW} \lesssim 1.147 \times 10^{53}$ erg in GWs and $1.591 \times 10^{53} \lesssim E_{other} < 2.484 \times 10^{53}$ erg are released in other energy forms. The energy observed in short GRBs and further theoretical analysis, including numerical simulations of the physical processes occurring during the merger, will clarify the efficiency of converting $E_{other}$ into observable radiation. Since no BH is formed (in this GM1 EOS analysis), the assumption that the merger leads to an S-GRF suggests an efficiency lower than 10%.

We now estimate the detection efficiency of the GW radiation released by the system in the post-merger phase when angular momentum losses are maximized, i.e., in the absence of a surrounding disk. We find the root-sum-squared strain of the signal, i.e.,

$$h_{rss} = \sqrt{\int 2\big[|\tilde{h}_+|^2 + |\tilde{h}_\times|^2\big]df} \approx \frac{1}{\pi d \bar{f}}\sqrt{\frac{G E_{GW}^{pm}}{c^3}}, \tag{14}$$

where $\tilde{h}_+$ and $\tilde{h}_\times$ are the Fourier transforms of the GW polarizations, $d$ is the distance to the source, $\bar{f}$ is the mean GW frequency in the postmerger phase. These signals are expected to be detected with a 50% of efficiency by the LIGO/Virgo pipelines [57] when $h_{rss} \sim 10^{-22}$ Hz$^{-1/2}$ [58]. For the energy release in the post-merger phase, we have $\bar{f} = 1671.77$ Hz, so these signals could be detected up to a distance of $d \approx 10$ Mpc.

## 5. Discussion and Conclusions

As some BdHN I and II systems remain bound after the GRB-SN event, the corresponding NS-BH and BNS systems, driven by GW radiation, will merge and lead to short GRBs. For a few minutes binary, the merger time is of the order of $10^4$ year. This implies that the binaries will still be close to the long GRB site by the merger time, which implies a direct link between long and short GRBs [20].

The occurrence rate of long and short bursts, however, should differ as the SN explosion likely disrupts the binaries with long orbital periods. We are updating our previous analysis on this interesting topic reported in [59]. We refer the reader to Bianco et al. [60] for a preliminary discussion.

As a proof of concept, this article examined this unique connection between long and short GRBs predicted by the BdHN scenario, emphasizing the case of mergers of BNS left by BdHNe II. For this particular case, the simulations predict that the outcome system will be a NSs binary with the star spins anti-aligned. The application of the present theoretical framework to the analysis of other merging binaries, such as the BH-NS binaries produced by BdHN I (see [20] for a general discussion), will be addressed in a separate work.

We have carried out a numerical SPH simulation of a BdHN II occurring in a CO-NS of orbital period 4.5 min. The mass of the CO is 3.06 $M_\odot$ and that of the NS companion, 1.4 $M_\odot$. The CO is the pre-SN star obtained from a ZAMS star of $M_{zams} = 15\, M_\odot$ simulated from MESA code. The SPH simulation follows [23,43]. It computes the accretion rate onto the $\nu$NS (left by the CO core collapse) and the NS companion while the ejecta expands within the binary. For the event that left a $\nu$NS-NS eccentric binary of $1.505 + 1.404\, M_\odot$, orbital separation $2 \times 10^{10}$ cm, orbital period of $\approx$15 min and eccentricity $e = 0.45$. The SN ejecta matter forms a disk around both stars with opposite spins, so we expect that the $\nu$-NS binary will also have anti-aligned spins as well. The above parameters suggest the BNS merger leading to a short GRB occurs in $\approx$73 kyear after the BdHN II event.

Whether or not the central remnant of the BNS merger will be a Kerr BH or a massive, fast-rotating NS depends on the nuclear EOS. For instance, we have shown the GM1 EOS leads to the latter while the TM1 EOS leads to the former. As an example of the theoretical framework presented in this article, we quantify the properties of the merger using the GM1

EOS. We infer the mass of the NS central remnant and the surrounding disk as a function of the angular momentum losses. We then emphasize the merger features in the limiting cases of maximum and zero angular momentum loss, corresponding to a surrounding disk's absence or maximum mass. We estimated the maximum energy and angular momentum losses in GWs. We showed that the post-merger phase could release up to $\approx 10^{52}$ erg in $\approx 1.7$ kHz GWs, and LIGO/Virgo could, in principle, detect such emissions for sources up to $\approx 10$ Mpc. We assessed that up to a few $10^{53}$ erg of energy could be released in other forms of energy, so a $\lesssim 10\%$ of efficiency of its conversion into observable electromagnetic radiation would lead to an S-GRF.

The direct link between long and short GRB progenitors predicted by the BdHN model opens the way to exciting astrophysical developments. For instance, the relative rate of BdHNe I and II and S-GRBs and S-GRFs might give crucial information on the nuclear EOS of NSs and the CO-NS parameters. At the same time, this information provides clues for the stellar evolution path of the binary progenitors leading to the CO-NS binaries of the BdHN scenario. Although challenging because of their expected ultrashort duration, observing a U-GRF would also be relevant for constraining the EOS of NS matter. An extended analysis is encouraged, including additional BNS parameters obtained from SPH simulations of BdHNe for various CO-NS systems and nuclear EOS.

**Author Contributions:** Conceptualization, L.M.B., C.F., J.F.R., J.A.R. and R.R.; methodology, L.M.B., J.F.R. and J.A.R.; formal analysis, L.M.B., J.F.R. and J.A.R.; investigation, L.M.B., J.F.R. and J.A.R.; writing—original draft, L.M.B., C.F., J.F.R., J.A.R. and R.R.; writing—review and editing, L.M.B., J.F.R. and J.A.R. All authors have read and agreed to the published version of the manuscript.

**Funding:** L.M.B. is supported by the Vicerrectoría de Investigación y Extensión—Universidad Industrial de Santander Postdoctoral Fellowship Program No. 2023000359. J.F.R. has received funding/support from the Patrimonio Autónomo—Fondo Nacional de Financiamiento para la Ciencia, la Tecnología y la Innovación Francisco José de Caldas (MINCIENCIAS–COLOMBIA) Grant No. 110685269447 RC-80740-465-2020, Project No. 69553.

**Data Availability Statement:** No new data were created or analyzed in this study. Data sharing is not applicable to this article.

**Conflicts of Interest:** The authors declare no conflict of interest.

## Abbreviations

The following abbreviations are used in this manuscript:

| | |
|---|---|
| BdHN | Binary-driven hypernova |
| BH | Black hole |
| BNS | Binary neutron star |
| CO | Carbon-oxygen |
| EOS | Equation of state |
| GRB | Gamma-ray burst |
| GW | Gravitational wave |
| ISCO | Innermost stable circular orbit |
| NS | Neutron star |
| $\nu$NS | Newborn neutron star |
| S-GRB | Short gamma-ray burst |
| S-GRF | Short gamma-ray flash |
| SN | Supernova |
| U-GRB | Ultrashort gamma-ray burst |
| U-GRF | Ultrashort gamma-ray flash |
| ZAMS | Zero-age main-sequence |

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
