# Peer review of "Neutron Star Binaries Produced by Binary-Driven Hypernovae, Their Mergers, and the Link between Long and Short GRBs"

_universe, doi:10.3390/universe9070332_

Round 1
Reviewer 1 Report
here is my referee report:
###
Referee report on "Neutron star binaries produced by binary-driven hypernovae, their
mergers, and the link between long and short GRBs" by L. M. Becerra et al.
This is a fascinating topic and is well done here. What might help these argument is the
observational result, that massive stars are very often in triple or quadruple systems. In a
quadruple system there are two massive stars in tight orbits, and the two tight binaries are
relatively far apart. That means that their two orbital spins could point in two different directions.
It would help to understand how these four stars can end, as second generation black hole mergers
are clearly observed. As all massive stars have substantial mass loss before they explode, the path
to get them close needs to be understood.
There are a few questions:
Why is the paragraph in the introduction starting with line 32 almost identical to the abstract?
That seems odd.
In line 112 it is stated that the ejected matter remains bound to the system. Why is that? That
needs justification.
In line 133 there is the statement "the angular momentum carried out by the ejecta since it is
expected to have small mass ∼ 10^{−4}–10^{−2} M_⊙ and to expand almost radially." That
seems inherently implausible. Remember, in the Solar system most of the angular momentum of
the system is in the planets and only a tiny fraction of the mass; there are similar masses.
Eq (4) assumes that the spins are all parallel. The LIGO/VIRGO/KAGRA data make clear that
this assumption is not warranted for black holes, and so why should that be the case for neutron
stars? Also, radio observations of super-massive black holes clearly show that the total spin before
the merger often has a different direction than that of the initially more active black hole, with
often very large angular differences, even in projection. So why should neutron star mergers be
simpler?
Did the authors redo the equations in Oppenheimer & Volkoff (1939 PRD) using large rotation
and with the Kerr metric to get eq (7)? That is not obvious here, and would really help to clarify.
In the summary or introduction it would help if the general context were also briefly summarized:
What happens for more massive stars, when there are black holes involved? Muxlow et al. (2005)
and others have argued that this could also lead to a GRB. What happens, when more than two
stars are involved, leading to either neutron stars or black holes? That is observationally a very
common case. Black holes as a result occur about 1/5 as often as neutron stars, and so constitute a
sizable fraction (Chieffi, Diehl, ...). Less reliance on acronyms would also help.
This is an interesting paper, but for the general reader more of the context would help.
here is my referee report:
###
Referee report on "Neutron star binaries produced by binary-driven hypernovae, their
mergers, and the link between long and short GRBs" by L. M. Becerra et al.
This is a fascinating topic and is well done here. What might help these argument is the
observational result, that massive stars are very often in triple or quadruple systems. In a
quadruple system there are two massive stars in tight orbits, and the two tight binaries are
relatively far apart. That means that their two orbital spins could point in two different directions.
It would help to understand how these four stars can end, as second generation black hole mergers
are clearly observed. As all massive stars have substantial mass loss before they explode, the path
to get them close needs to be understood.
There are a few questions:
Why is the paragraph in the introduction starting with line 32 almost identical to the abstract?
That seems odd.
In line 112 it is stated that the ejected matter remains bound to the system. Why is that? That
needs justification.
In line 133 there is the statement "the angular momentum carried out by the ejecta since it is
expected to have small mass ∼ 10^{−4}–10^{−2} M_⊙ and to expand almost radially." That
seems inherently implausible. Remember, in the Solar system most of the angular momentum of
the system is in the planets and only a tiny fraction of the mass; there are similar masses.
Eq (4) assumes that the spins are all parallel. The LIGO/VIRGO/KAGRA data make clear that
this assumption is not warranted for black holes, and so why should that be the case for neutron
stars? Also, radio observations of super-massive black holes clearly show that the total spin before
the merger often has a different direction than that of the initially more active black hole, with
often very large angular differences, even in projection. So why should neutron star mergers be
simpler?
Did the authors redo the equations in Oppenheimer & Volkoff (1939 PRD) using large rotation
and with the Kerr metric to get eq (7)? That is not obvious here, and would really help to clarify.
In the summary or introduction it would help if the general context were also briefly summarized:
What happens for more massive stars, when there are black holes involved? Muxlow et al. (2005)
and others have argued that this could also lead to a GRB. What happens, when more than two
stars are involved, leading to either neutron stars or black holes? That is observationally a very
common case. Black holes as a result occur about 1/5 as often as neutron stars, and so constitute a
sizable fraction (Chieffi, Diehl, ...). Less reliance on acronyms would also help.
This is an interesting paper, but for the general reader more of the context would help.
Reviewer 2 Report
The authors discuss a model of origin and evolution of gamma-ray bursts (GRBs). GRBs are extremely energetic events which can produce multiwavelength electromagnetic emission, gravitational radiation, and neutrino signals observable from cosmological distances. They provide important information on the nature of most energetic astrophysical phenomena occurring in different places of the Universe. Accordingly, they are valuable for cosmology, and many aspects of fundamental physics and astrophysics.
The authors develop the alternative concept of GRBs based on the model of binary-driven hypernovae (BdHNe). The model is not standard, but different models are currently welcome, taking into account evident complexity of the GRB phenomena. The present team has started to develop the BdHN model in previous publications. Here the authors describe their classification of BdHN types, and present original numerical simulations supplemented by a simi-analytic analysis, making the concept more elaborated.
I have no serious questions/comments, but I can formulate a few minor remarks which might help improving presentation of the material. Since the remarks are minor, I recommend the paper for publication after the authors are acquainted with these remarks and implement them if they find them useful.
Here is the list:
Lines 4 and 5. Instead of “The SN accretes on the nuNS and the NS” it can be more accurate to write “The nuNS and the NS accrete SN matter,” or something like this.
The next sentence. Instead of “BdHNe classifies…” I would suggest “BdHNe are classified…”
Page 2. Instead of introducing the abbreviation “BNS” on line 77, it would better to introduce it earlier (e.g., on line 46?), where it is first mentioned in the text.
I suggest to move the wording “peak energy” from line 60 to line 52, for explaining the meaning of “E_p,i” there.
There are too many abbreviations throughout the text, and the reader can be lost. I recommend to create a table of the main abbreviations with explanations in the end of the paper (as encouraged by the rules of Universe). The reader will be grateful.
Fig. 1. What does the notation “x [x 1e-1 Rsun]” mean? If it is “x/(0.1 Rsun),” it would be better to write this explicitly. The same refers to the vertical axis.
Table 1. Does the first column show label of the object “i”? If yes, it would be better to insert “i” in the first line and explain in the text. The third column presents dimensionless angular momentum (as explained on line 102). If it is dimensionless, no dimension should be indicated in the table. In addition, I suggest explaining SPH (smoothed-particle hydrodynamics) in the caption, for clarity.
Line 104. The names GM1 and TM1 of the adopted EOSs are not very informative. Can the authors present at least maximum gravitational masses of NSs for these EOSs?
Line 125. It would be better to replace comma by semicolon before “J_i”.
Line 129. I would replace “momentum” by “momenta”
Line 131. I would again replace “momentum” by “momenta”
Line 132. Replace “account” with “accounts”
Line 146. Abbreviation “ISCO” needs to be explained
Line 157. Please, replace “which below” with “below which”
Eq. (10). The two versions, for q \appox 1 and q<1, do not match at q=1. Is it true? If yes, what is the expected behavior as q tends to 1?
Caption to Fig. 2. Line 4. Would it better to add “it” after “TM1 EOS”
Line 192. Add “that” after “shows”
Line 202. I would replace “erg” with “ergs” to be consistent with “are carried out”. The same applies to lines 222 and 226.
Line 234. Replace “transform” with “transforms”
Line 237. Add “we have” after “post-merger phase,”
Line 249. The sentence starting with “The event left…” sounds unclear. Perhaps, it should be “For the event that left…”?
Reviewer 3 Report
The paper presents a detailed and enjoyable implications of the binary-driven hypernova (BdHN) model for interpretations of the long gamma-ray bursts (GRBs). In particular, the authors consider scenarios corresponding to a binary composed of a pre-supernova (SN) carbon-oxygen (CO) star and a neutron star (NS). The core collapse of CO-star and subsequent accretion on neutron stars creates a GRB that depends on hypernova parameters. The authors demonstrate that such a model provides realistic description of various GRBs.
The paper materials are interesting, contain results relevant for studies the nuclear equation of state and warrant to express publication in J.‘Universe’.
However, before recommending for publication, I suggest minor corrections.
|
It is worthy to present maximal neutron star masses for considered
equation of states
The authors should give more detailed discussion of magnetars since
respective magnetic field can affect accretion.
|
reasonable
Round 2
Reviewer 1 Report
There is only one minor additional change, that might help a reader: In the discussion the authors should explicitly state, as they do in the cover letter, why they expect the spins to be aligned, or anti-aligned. That is an interesting non-trivial point.
Other than this additional change, the manuscript is quite interesting and useful for a reader, and can be published after this change following their own cover letter.
There is only one minor additional change, that might help a reader: In the discussion the authors should explicitly state, as they do in the cover letter, why they expect the spins to be aligned, or anti-aligned. That is an interesting non-trivial point.
Other than this additional change, the manuscript is quite interesting and useful for a reader, and can be published after this change following their own cover letter.
